# Fat and Protein Combat Triggers Immunological Weapons of Innate and Adaptive Immune Systems to Launch Neuroinflammation in Parkinson’s Disease

**DOI:** 10.3390/ijms23031089

**Published:** 2022-01-19

**Authors:** Shelby Loraine Hatton, Manoj Kumar Pandey

**Affiliations:** 1Cincinnati Children’s Hospital Medical Center, Division of Human Genetics, 3333 Burnet Avenue, Cincinnati, OH 45229, USA; hattonsl@mail.uc.edu; 2Department of Pediatrics, Division of Human Genetics, College of Medicine, University of Cincinnati, 3333 Burnet Avenue, Cincinnati, OH 45229, USA

**Keywords:** lipid, alpha-synucleinopathy, inflammation

## Abstract

Parkinson’s disease (PD) is the second-most common neurodegenerative disease in the world, affecting up to 10 million people. This disease mainly happens due to the loss of dopaminergic neurons accountable for memory and motor function. Partial glucocerebrosidase enzyme deficiency and the resultant excess accumulation of glycosphingolipids and alpha-synuclein (α-syn) aggregation have been linked to predominant risk factors that lead to neurodegeneration and memory and motor defects in PD, with known and unknown causes. An increasing body of evidence uncovers the role of several other lipids and their association with α-syn aggregation, which activates the innate and adaptive immune system and sparks brain inflammation in PD. Here, we review the emerging role of a number of lipids, i.e., triglyceride (TG), diglycerides (DG), glycerophosphoethanolamines (GPE), polyunsaturated fatty acids (PUFA), sphingolipids, gangliosides, glycerophospholipids (GPL), and cholesterols, and their connection with α-syn aggregation as well as the induction of innate and adaptive immune reactions that trigger neuroinflammation in PD.

## 1. Introduction

Parkinson’s Disease (PD) is an age-related neurodegenerative disease that affects ~1-2/1000 people [1]. The prevalence of PD in the U.S. is ~572/100,000 people above the age of 45. Based on U.S. Census Bureau population records in 2010, the number of cases for those over the age of 45 in 2010 was 680,000, which is expected to rise to 1,238,000 cases by 2030 [2]. PD mainly happens due to loss of dopaminergic neurons in a very specific region of the brain—termed the substantia nigra pars compacta—specifically due to Lewy body formation, overactivation of the immune system, microgliosis, and oxidative stress [3,4,5,6,7,8,9,10,11,12,13,14]. These proinflammatory environments influence disease processing and the development of the motor (e.g., tremor, rigidity, bradykinesia/akinesia, and postural instability) [15,16,17] and non-motor symptoms, (e.g., changes in apathy, anhedonia, depression, cognitive dysfunction hallucinosis, and complex behavioral disorders) [17]. While 5–10% of PD cases are familial and happen due to several genetic defects [18], 90–95% of cases have unknown causes and are idiopathic in nature [19]. Glucocerebrosidase (GCase) deficiency and the resultant excess brain accumulation of glycosphingolipids, α-synuclein (α-syn) aggregation, proinflammatory cytokine generation, and reactive oxygen species (ROS) production have been linked to PD with both known and unknown causes [20,21,22,23,24,25,26,27,28,29,30,31]. Several other lipids have also been shown to have a direct or indirect role in the induction of α-syn aggregation and the massive generation of ROS and proinflammatory cytokines in PD [32,33,34,35,36]. α-syn is a cytosolic protein, found mostly in the brain, that is encoded by the SNCA gene [37]. It was discovered along with β-syn and γ-syn in the presynaptic terminals of axons [38]. Normal α-syn has been linked to the restriction of synaptic vesicle mobility, attenuation of synaptic vesicle recycling, stabilization of soluble N-ethylmaleimide-sensitive factor attachment protein receptor complexes via chaperone activity, and controlling neurotransmitter release [39,40,41,42]. However, the exact function of α-syn remains largely unknown [43]. α-syn at the monomeric level is defined as a soluble monomer with a disorganized confirmation [44]. α-syn accumulates in the form of Lewy bodies (LBs) and Lewy neurites (LNs) in a group of neurodegenerative diseases collectively known as synucleinopathies, which comprise PD, dementia with Lewy bodies (DLB), and multiple system atrophy (MSA) [45,46]. LBs and LNs accumulate α-syn and trigger its prion-like spreading [47,48]. Glycophosphatidylinositol (GPI) anchors, phosphatidylethanolamine (PE), gangliosides, and several lipid-like molecules have been linked to the conversion of normal cellular protein to pathogenic forms of prion proteins [49,50,51,52,53,54,55,56]. Aggregated forms of α-syn are able to recruit and seed the endogenous protein and initiate its spreading throughout the cells of the nervous system in animal and human models of synucleinopathies, aging, and and other brain diseases [57,58,59,60,61,62,63,64,65,66,67]. There is growing evidence indicating that abnormal α-syn can spread to neighboring brain regions and cause aggregation of endogenous α-syn in these regions as seeds, in a “prion-like” manner [68]. α-syn has a high affinity to negatively charged lipids, and therefore its lipid-binding domain also interacts with platelet-activating factor, glycerophosphocholine (GPL), and lipid species with shorter hydrocarbon tails, which all show a link to α-syn aggregation [69,70]. Furthermore, α-syn gene (SNCA) defects also altered α-syn lipid-binding domains, which suggests the importance of phospholipid membranes and monomeric α-syn interactions [71]. At the monomeric level, several of the lipids, such as triglyceride (TG), diglycerides (DG), glycerophosphoethanolamine (GPE), phosphatidic acid (PA), phosphatidylethanolamine (PE), and phosphatidylcholine (PC) have been associated with α-syn aggregation-mediated pathophysiology of PD, DLB, and Alzheimer’s diseases [72].

The α-syn aggregates have been linked more precisely to the activation of an innate and adaptive immune system that includes microglial cell activation, proinflammatory cytokine generation, and the loss of dopaminergic neurons in the substantia nigra pars compacta region of the brain [73,74]. Studies have also shown α-syn to be a key player in lipid metabolism regulation, dopamine production, and inflammatory responses [75,76,77,78,79,80,81,82]. α-syn activity in these roles is dependent on the binding and attraction of α-syn to negatively charged lipids such as GPE, TAG, DAG, GPI, cardiolipin (CL), docosahexaenoic acid (DHA), gangliosides, and other acidic phospholipids that influence the conformational changes and catalysis of the formation of abnormal species of α-syn, which lead to the activation of immune inflammation and neurodegeneration in PD [83,84,85,86,87]. Overall, the improper metabolism of several lipids triggers the abnormal formation of α-syn, which propagates brain inflammation in PD and other alphasynucleinopathies; however, this aspect has not received much focus in the field. Therefore, the underlying discussion in this paper explored several lipids and their link to α-syn abnormalities, and the induction of a variety of innate and adaptive immune players and their reactions that spark neuroinflammation in PD. Understanding the relationship between these biochemicals and their impact on the induction of neuroimmune inflammation may reveal alternative treatments and therapies for PD.

## 2. Triglyceride-Induced α-syn Abnormalities in PD

Triglyceride (TG) is also known as triacylglycerol (TAG), which is an ester derived from glycerol and three fatty acids. TAG storage primarily occurs in hepatocytes. Under normal circumstances on a daily basis, the liver causes the processing of large quantities of fatty acids but stores only small amounts in the form of TAG [88]. This is because the rates of acquisition of FA—by uptake from the plasma and from de novo synthesis within the liver—are balanced by rates of FA oxidation and secretion into plasma as TG-enriched, very low-density lipoprotein (VLDL-TG). The relatively small quantities of TAG stored within the liver are localized in cytoplasmic lipid droplets, which provide energy to the biological system. 

Hypertriglyceridemia results from the accumulation of TG-rich lipoproteins (e.g., VLDL, VLDL remnants, and chylomicrons) in the bloodstream. This condition is very often the result of genetic and environmental factors, but can also result from specific genetic disorders, such as hypertriglyceridemia, familial combined hyperlipidemia, familial dysbetalipoproteinemia, familial chylomicronemia, and aging [89,90]. Excess TAG making and storage have been observed in obesity, arteriosclerosis, strokes, heart attacks, pancreatitis, and preeclampsia [91,92,93,94,95,96]. α-syn accumulation in response to lipid droplets with high levels of TAG has been observed in many studies [97]. In HeLa and hippocampal neurons of certain A53T mutants, abnormal hydrolysis of stored TAG was associated with increased intracellular accumulation and enhanced activity of the acyl-CoA synthetase enzyme, which causes the formation of acyl-CoA and beta-oxidation responsible for biosynthesis of glycerophospholipids and sphingolipids [97]. Conversely, an increase in mutant α-syn leads to an increase in TAG hydrolysis, and consequently a decrease in intracellular TAG in A53T and A30P transgenic mice and of PD patients, which suggests a link between abnormal lipid metabolism and α-syn aggregation [98,99,100]. Low levels of TAGs in serum and plasma of mouse models and human patients with PD (Table 1 and Table 2) and their link to α-syn accumulation and aggregation have been observed in many studies [97,98,99,100], which suggests that TAG-induced α-syn abnormalities process the disease in PD. 

## 3. Glycerophosphoethanolamine (GPE)-Induced α-syn Abnormalities in PD

Glycerophosphoethanolamine (GPE) is involved in cell membrane metabolism [101,102]. However, higher levels of GPE have been linked to liver cirrhosis and Alzheimer’s disease [101,103]. GPE has been shown to play a role in enhancing the preferential binding of acidic phospholipids to α-syn, and the resultant increases in α-syn aggregation concluded that GPE is vital for the induction of α-syn abnormalities in PD [43]. GPE has also been associated with the formation of ion channels in monomeric wild-type, E46K mutant, and A53T mutant α-syn models. The monomeric form of wild-type α-syn and its mutants formed ion channels with membranes containing 50% GPE, while oligomeric α-syn lacked such formation of ion channels and PE requirements. These observations indicate that interactions between monomeric α-syn and GPE are critical for the abnormal development of α-syn in PD [104].

## 4. Diglyceride (DG)-Induced α-syn Abnormalities in PD

Diglyceride (DG), or diacylglycerol (DAG), is a class of lipid, which is composed of two fatty acids chains covalently bonded to one glycerol through ester linkages. Dietary fat is mainly composed of TAG since it cannot be absorbed by the digestive system, hence TAG must first be enzymatically digested into monoacylglycerol (MAG), diacylglycerol (DAG), or free fatty acids. DAG is a precursor to TAG, which is formed by the addition of a third fatty acid to the DAG under the catalysis of diglyceride acyltransferase. DAG can be phosphorylated to phosphatidic acid by diacylglycerol kinase. DAG synthesis involves the hydrolysis of the phospholipid phosphatidylinositol 4, 5-bisphosphate (PIP_2_) by the enzyme phospholipase C (PLC) (a membrane-bound enzyme), which through the same reaction produces inositol trisphosphate (IP_3_). Although IP3 diffuses into the cytosol, DAG remains within the plasma membrane due to its hydrophobic properties. IP_3_ stimulates the release of calcium ions from the smooth endoplasmic reticulum, whereas DAG is a physiological activator of protein kinase C (PKC) [105,106]. Increased intracellular DAG-induced activation of PKC has shown decreased insulin-stimulated IRS-1/IRS-2 tyrosine phosphorylation, PI3K activation and downstream insulin signaling in liver and skeletal muscles [107]. The DAG kinase enzyme converts DAG to PA, while the DAG acyltransferase enzyme is responsible for the conversion of DAG to TAG [108]. Studies have shown that a strong binding of 18:1/18:1 species of PA to α-syn causes its aggregation, mainly due to high DAG and low PA levels in PD [109,110]. These findings suggest that the abnormal interaction between the DAG–PA axis triggers α-syn aggregation in PD.

## 5. Polyunsaturated-Fatty-Acid (PUFA)-Induced α-syn Abnormalities in PD

Polyunsaturated fatty acid (PUFA) is a fatty acid that contains more than one double bond in the glycerol backbone; on the basis of their carbon backbone length, they are classified in two groups, i.e., short-chain PUFA (SC-PUFA), with 18 carbon atoms; and long-chain PUFA (LC-PUFA), with 20 or more carbon atoms [111]. Several such PUFA, including ω-6 (or *n*-6), arachidonic acid (ARA; 20:4*n*-6), and ω-3 (or *n*-3), eicosapentaenoic acid (EPA; 20:5*n*-3), and docosahexaenoic acid (DHA; 22:6*n*-3) were reported to be essential for (1) neuronal membrane and visual brain development, (2) maintenance of neuronal membrane fluidity and permeability, (3) controlling of cardiovascular and coronary artery function, and (4) regulation of inflammation [112,113,114,115,116,117,118,119,120,121,122]. The abnormal ratio between dietary ω-6 (n-6) to ω-3 (n-3) PUFA has been linked to the pathophysiology of cancer, rheumatoid arthritis, atherosclerosis, and obesity [123,124,125].

α-syn and PUFA are involved in synaptic vesicle recycling [126]. However, prolonged interaction of PUFA and monomeric α-syn causes α-syn fibril formation and its oligomerization, which supports the formation of Lewy-like proteinaceous bodies in the cytoplasm [81,127,128,129,130,131,132,133,134,135]. Interaction of α-syn with 4-hydroxy-2-nonenal, a product of PUFA peroxidation, seems to produce the same result, but was also found to prevent the passage of fibrils into Lewy Bodies [127,129,130]. PUFA levels have been found to be altered in many models of PD. Several PUFA, specifically linoleic acid (LA), docosatetraenoic acid (DSA), eicosapentaenoic acid (EPA), and osbond acid (OBA), were elevated in mesencephalic (MES) neuronal cells of PD patients [135]. In mice models, levels of LA, EPA, and eicosatrienoic acid (ESA) were also elevated in MES neurons [81] and PD brains [135]. Two PUFAs of special note in α-syn interactions are docosahexaenoic acid (DHA) and arachidonic acid (ARA). α-syn levels contribute to an increase in ARA and DHA81. Transmission electron microscopy, electrophoresis of native gels, and fluorescence assays with thioflavin indicate that DHA and ARA trigger α-syn oligomerization and cause an immediate transformation of α-syn to its helical conformation [136,137]. Coincidentally, other studies revealed that α-syn has a role in the addition of substrates to acyl-CoA synthetase. As a consequence, α-syn’s addition of substrates allows control of the brain level of PUFA [138]. These observations indicate that α-syn and PUFAs co-regulate each other via complex interactions [81]. Prolonged exposure of DHA to α-syn causes α-syn oligomerization by activating the retinoic X receptor and peroxisome proliferator-activated receptor gamma 2 [139], therefore resulting in the production of lipid droplets [137]. Furthermore, these lipid droplets could be remodeled by α-syn and increase assemblage of amyloid fibrils that contain DHA [136]. Accordingly, levels of DHA and ARA were higher in the frontal cortex of PD patients. Conversely, their levels appeared lower in frontal cortex lipid rafts [140]. In mice models, ARA levels were elevated in the MES neurons [135], frontal cortex, striatum, midbrain synaptosomes, and striatal synaptosomes [141,142]. Furthermore, DHA was found to be elevated in MES DA neuronal cells and brains of PD patients, and it was concluded that increased DHA intake causes increased α-syn toxicity [135]. A possible mechanism for this is the disruption of membrane integrity via leakage of small molecules out of the vesicles due to oligomeric α-syn binding [143]. Interestingly, oligodendroglia cells that expressed both the wild-type and mutated A53T α-syn resulted in fibril formation, thus solidifying the conclusion that elevated DHA affects α-syn fibril formation [144]. Increased levels of several species of PUFA, i.e., αLA, EPA, ETA, and ARA in neurons of mouse models (Table 1) and DHA, LA, DSA, EPA, and OBA in neuronal cells and brains of human PD patients (Table 2), as well as their relation to α-syn oligomerization, have been observed [81,127,128,129,130,131,132,133,134,135]. These findings suggest that several species of PUFA could develop α-syn abnormalities in PD.

## 6. Sphingolipid-Induced α-syn Abnormalities in PD

Sphingolipids are molecules with lipid chains forming part of the cell lipid membrane and have been identified as inflammatory regulators of cellular pathways [145,146,147]. Sphingolipid metabolism typically begins with the production of ceramides, which are the major sphingolipids released during cellular metabolism, responsible for the degradation of sphingosine [148]. Such sphingosine is in turn phosphorylated by the isoenzymes sphingosine kinase 1 (SphK1) and sphingosine kinase 2 (SphK2) into sphingosine-1-phosphate (S1P) [149,150]. This specific pathway regulates cell growth, apoptosis, survival, and inflammation in several diseases, such as systemic lupus erythematosus, cancer, coronary artery disease, and the fibrotic diseases of the lungs, liver, and heart [151,152,153,154,155,156,157,158,159,160]. Abnormal synthesis of ceramides, sphingosine, and S1P have been linked to α-syn aggregation in PD [161]. One source of ceramides and sphingosine for this pathway comes from the hydrolysis of glucosylceramide and glycosylsphingosine via acid β-glucosidase. Interestingly, mutations in the *GBA1/Gba1* gene that encodes the enzyme acid β-glucosidase has been strongly associated with the development of PD, DLB, and lysosomal storage disease, (e.g., Gaucher disease) [18,162,163,164,165,166,167]. Studies have also shown the connection between lower levels of ceramides and α-syn aggregation in PD patients [110,168,169,170,171]. Analysis of brains from postmortem PD patients revealed a direct relationship between increased glycosylsphingosine and α-syn levels due to a lack of glycosylsphingosine hydrolysis into ceramides, which revealed an inverse relationship between such glycosylsphingosine and α-syn levels in the brain [172]. These findings suggested that ceramide conversion to sphingosine seems to be a crucial step for α-syn aggregation. Higher levels of sphingosines were associated with the formation of oligomeric α-syn, which is attributed to fibril formation. Further conversion to S1P due to GBA deficiency also contributed to α-syn aggregation [173]. Interestingly, increased levels of α-syn inhibited phosphorylation of sphingosine to S1P and affected S1P-receptor-mediated signaling. Conversely, levels of ceramides, specifically monohexosylceramides, were found to be elevated in PD patients with *GBA* mutations. Accordingly, higher ceramide levels have been observed in the plasma, primary visual cortex, and CSF of PD patients [168,173,174,175]. This trend has also been observed in mice models of Lrrk2 and PINK1 [176,177]. Other sphingolipids, such as sphingomyelin (SM) and cerebrosides, have also had altered levels in PD models; for example, higher levels of SM have been found in plasma [175], the primary visual cortex [178] and ACC of humans with PD [168]. Elevated levels of SM and cerebrosides have been observed in substantia nigra of male PD patients [179]. Interestingly, SM was also observed to be lower in the ACC of PD [168] patients and the CSF of AD patients [174]. SM treatment of SK-N-SH neuronal cells increased the expression of α-syn. One proposed pathway is that SM acts as a substrate for the ABCA5 transporter [180]. Although the mechanism remains unknown, higher levels of SM resulted in higher SM transportation and α-syn accumulation, therefore linking SM metabolism to the pathogenesis of many LBDs. Cerebroside levels were observed to be higher and lower in the plasma [175] and frontal cortex, respectively [140]. Furthermore, decreased levels of cerebroside have been linked to increased levels of ceramides, therefore contributing to the formation of fibrils and aggregates [181]. Elevated neurons, olfactory bulbs, and brain levels of sphingolipids, i.e., sphingosine and ceramide, have been observed in mouse models of PD (Table 1). Similarly, several members of the sphingolipids, including sphingosine, SM, ceramide, cerebroside, and glucosylceramide have been observed in the plasma, fibroblast, neurons, frontal cortex, anterior cingulate cortex (ACC), primary visual cortex (PVC), substantia nigra, and other brain regions of human patients with PD (Table 2). Several such sphingolipids have been linked to α-syn aggregation in PD [110,161,168,169,170,171], suggesting the importance of sphingolipids in the propagation of α-syn-mediated pathophysiology in PD.

## 7. Ganglioside-Induced α-syn Abnormalities in PD

Ganglioside (GAG) is an important component of the neuronal cell membrane, which is composed of a ceramide lipid tail attached through glycosidic linkage to a glycan headgroup containing one or more sialic acid residues (e.g., n-acetylneuraminic acid) [182]. GAG is functionally involved in the formation of the neuronal membrane and conduction of neurotransmission processing [183,184,185,186]. GAG is synthesized in the Golgi apparatus and trafficked to different parts of the cells, such as the plasma membrane [187,188], endo-lysosomal system [189,190], endoplasmic reticulum (ER)–mitochondria contact sites [191,192], and the nuclear envelope [193,194]. In addition to PD, GAG deficiency has been also linked to several brain diseases, such as aging, neurodevelopmental diseases, Alzheimer’s, and Huntington’s disease [195,196,197,198,199,200,201,202,203,204,205,206,207,208,209,210].

α-syn was determined to have a GM3-GAG-binding domain, and to therefore preferentially bind to GM3 [211]. Interestingly, this particular domain was found to resemble a domain responsible for the binding of amyloid proteins, which includes the binding of β-amyloid peptide, which forms amyloid plaques in Alzheimer’s disease [211,212]. Decreased levels of GM1-GAG have been observed in the putamen [210] and substantia nigra of PD patients [210,213]. Elevated levels of GM2-GAG have been found in fibroblasts of PD patients [214]. Similarly, elevated levels of GM3-GAG were observed in plasma, cerebrospinal fluid, and brain regions of PD patients [210,215,216,217]. Elevated levels of several of the gangliosides, (e.g., GM1, GM2, and GM3) have been observed in plasma, fibroblasts, SN, and brains of human patients with PD (Table 2). Altered levels of the GM1-GM2-GM3 axis have been linked to α-syn aggregation PD [212,218,219,220]. These findings suggest the importance of the GM1-GM2-GM3 axis in the induction of α-syn aggregation in PD.

## 8. Glycerophospholipid-Induced α-syn Abnormalities in PD

Glycerophospholipids (GPL), or phosphoglycerides, are glycerol-based phospholipids that comprise at least one *O*-acyl residue attached to the glycerol moiety. Several GPL species, such as phosphatidylcholine (PC), phosphatidylethanolamine (PE), and phosphatidylserine (PS) are abundant in mammalian cell membranes, which are critical for the formation of the cerebral structure and the steering of cognitive functions [221,222]. Alterations in GPL composition were observed in several brain diseases, such as schizophrenia, Huntington’s, and Alzheimer’s disease [222,223,224,225].

α-syn shows preferential binding to negatively charged acidic lipids [87]. Due to the negative charge, its lysine residues become seamless acnes for α-syn binding [226]. These findings suggest that α-syn binding to PS and phosphoinositols (PIs), the most acidic forms of phospholipids, leads to the formation of α-syn oligomers [227]. The phosphatidylinositol 4,5-bisphosphate is also the phosphorylated form of PI, and is termed as a phosphoinositide, which has also shown its tendency to interact with α- and γ-syn and cause their aggregation [228]. The binding of such PI seems to mostly occur in the presynaptic terminals of vesicles that show the presence of α-syn, indicating the possible role of PI in α-syn abnormalities [228,229]. Lower levels of PI and PS have been observed in the substantia nigra of PD patients [179]. Accordingly, α-syn knockout mice showed increased PI and PS acylation [230], and α-syn-deficient models of PD revealed an increase in DHA conversion to PI and PS [87]. Other studies have shown elevated levels of PS in fibroblasts, the frontal cortex, and the primary visual cortex [110,178,214].

Phosphatidic acid (PA) is secondary lipid messenger that, as with PI, has roles in intracellular trafficking of vesicles [231]. Due to its acidic nature, PA has been found to preferentially bind to α-syn residuals 1 to 102 [232]. Elevated levels of PA have been found in the plasma of PD patients [231,233]. Likewise, phosphatidylglycerol (PG) in micellar tubules has been shown to mitigate amyloid formation by membrane remodeling [234]. Along with PI and PS levels, PG levels are elevated in the frontal cortex of PD, AD, and LBD patients, thus highlighting its role in fibril formation in Lewy body disorders. Unlike the other glycerophospholipids, PE has recorded lower levels of expression in plasma, the visual cortex, and substantia nigra, and higher levels in the frontal cortex of PD models and patients [140,178,235,236]. PE was also found to aid in lipid vesicle binding to α-syn by creating a monolayer curvature strain on these vesicles, and as such, α-syn binding of them [43]. Altered levels of glycerophospholipids, i.e., CL in embryonic fibroblasts of mouse models of PD (Table 1), as well as PA, PC, PE, PS, PI, PG, and APG, have been found in the plasma, frontal cortex, visual cortex, fibroblasts, SN, and brains of human patients with PD (Table 2). Several such glycerophospholipids have been associated with the formation of α-syn oligomers [227,229]. These findings suggest that abnormal chemical reactions between glycerophospholipids and normal α-syn promote α-syn oligomerization in PD. 

## 9. Cholesterol-Induced α-syn Abnormalities in PD

Cholesterol is essential for cell membrane physiology, dietary nutrient absorption, reproductive biology, stress responses, salt and water balance, calcium metabolism, proper functioning of oncogenic G proteins, proteases of amyloid precursor protein, synapse formation, and synthesis of steroid hormones, bile acids, and vitamin D [237,238,239,240,241]. The human body produces around one gram of cholesterol/day [242]. Synthesis of cholesterol is a series process that starts with acetyl CoA and acetoacetyl-CoA, which are hydrated to form 3-hydroxy-3-methylglutaryl CoA (HMG-CoA), and subsequently reduced to mevalonate by HMG-CoA reductase enzyme [243]. Both dietary and synthesized de novo cholesterols are transported by lipoprotein particles through the circulatory system. The four major types of lipoproteins are chylomicron, very low-density lipoprotein (VLDL), low-density lipoprotein (LDL), and high-density lipoprotein (HDL), where chylomicrons and VLDL deliver TAG to cells, LDL delivers cholesterol to cells, and HDL is involved in reverse cholesterol transport [244,245].

The synthesis and utilization of cholesterol is a strongly controlled method that prevents its overdeposition of tissue. However, abnormal synthesis and utilization of cholesterol due to certain inherited diseases, infection, and/or unhealthy lifestyle, contributes to the development of hyperlipidemia and hypolipidemia [246,247,248,249,250]. Several illnesses, such as atherosclerosis, xanthomas, tangier, strokes, and cardiovascular diseases have been linked to high cholesterol levels [251,252,253,254]. Meanwhile, low cholesterol levels have been linked to cognitive defects (e.g., depression, suicide, anxiety, impulsivity, and aggression) in several brain sicknesses, such as trauma, hemorrhagic stroke, and Huntington’s disease [255,256,257,258,259,260,261].

Studies have linked cholesterol abnormalities to α-syn aggregation; for example, PARKN expression is one of the major dysfunctional genes in PD, which is controlled by cholesterol metabolism through ubiquitination of CD36, and in turn regulates the production of cholesterol [262]. Such cholesterol interaction has normal α-syn accumulation and aggregation [263,264]. It was also demonstrated that decreased levels of cholesterol reduced the amount of α-syn and mitigated the effects of its aggregation [265]. The exact role of cholesterol in the pathogenesis of PD is controversial, as human studies have found cholesterol levels to be both high [266] and low in plasma samples of PD patients [267]. Some studies showed that lower levels of cholesterol increased the risk of PD, while higher levels reduced the risk of PD [268,269]. Moreover, other studies showed no correlation between cholesterol levels or intake and PD risk [270]. In mice models, cholesterol levels were observed to be elevated in MEF cells [271] and lower in serum [272]. Interestingly, cholesterol levels have been found to be both high and low in mice astrocytes [77,273] and fibroblast [271,273]. Overall, altered levels of cholesterol have been found in serum, fibroblasts, astrocytes, and MES. cells of mouse models of PD (Table 1), and abnormal levels of such cholesterol have been observed in the plasma and PVC of human patients with PD (Table 2). Overall, abnormal cholesterol production and its interaction to normal α-syn caused α-syn aggregation in both PD mouse models and patients [262,263,264,265]. These findings suggest that altered biochemistry of cholesterol and its linking to α-syn subsidizes α-syn aggregation in PD.

**Table 1 ijms-23-01089-t001:** Lipids linked to α-syn abnormalities in mouse model of PD.

Family	Types	PD Mouse Model’s Alpha-Synuclein Abnormalities	Source	Levels	Reference
Glycerolipid	TAG	*A53*T mutant	Serum	Low	[272]
	NEFA	*A53*T mutant	Serum	Low	[272]
PUFA	α-LA	Transgeneic alpha-synuclein mice	MES Neurons	Induces	[81]
		alpha-synuclein null C57Bl6 background	MES Neurons	High	[135]
	EPA	Transgeneic alpha-synuclein mice	MES Neurons	Induces	[81]
	ETA	a-syn null C57Bl6 background	MES Neurons	High	[135]
	ARA	alpha-synuclein null C57Bl6 background	MES Neurons	High	[135]
		*Fabp3*KO mice with MPTP treatment	PC12 cells	High	[274]
MUFA	OA	*A53*T overexpressing cells	MES Neurons	Low	[135]
Glycerophospholipid	CL	*PINK1* KO	Embryonic fibroblasts	Low	[275]
Sphingolipids	Sphingosine	*GbaL444P*/KO *SNCA*T and *Gba*KO/KO *SNCA*T	Neuron	Induce	[173]
	Ceramide	*Lrrk2*KO mice	Brain	High	[176]
		*PINK1* KO	Olfactory bulb	High	[177]
Sterol	Cholesterol	*DJ1* KO	Embryonic fibroblasts	Low	[273]
		*DJ1* KO	Astrocyte	Low	[273]
		*GBA1* KO	Fibroblasts	Low	[271]
		*PRKN* KO	MEF cells	High	[276]
		*A53*T mutant	Serum	Low	[272]
		alpha-synuclein gene-ablated mice	Astrocyte	High	[77]

Polyunsaturated fats (PUFA), monounsaturated fats (MUFA), triacylglycerols (TAGs), non-esterified fatty acids, α-linoleic acid (α-LA), eicosapentaenoic acid (EPA), eicosatetraenoic acid (ETA), arachidonic acid (ARA), oleic acid (OA), cardiolipin (CL), alpha-synuclein (a syn), knockout (KO), T (Tansgenic),.and mesencephalic (MES).

**Table 2 ijms-23-01089-t002:** Lipids linked to α-syn abnormalities in human patients with PD.

Family	Types	Source	Level	Reference
Glycerolipid	TAG	Plasma	Low	[217,236,277]
PUFA	DHA	MES	High	[135]
		Brain	High	[135]
	LA	MES	High	[135]
		Brain	Low	[135]
	DSA	MES	High	[135]
		Brain	High	[135]
	EPA	MES	High	[135]
	OBA	MES	High	[135]
Eicosanoids	PGL J2	SK-N-SH cells	Induces	[278]
Glycerophospholipids	PA	Plasma	Low	[235]
	PC	Frontal cortex	Low	[110]
	PE	SN	Low	[235]
		Visual cortex	Low	[178]
		Plasma	Low	[236]
		Frontal cortex	High	[140]
		Plasma	Low	[175]
	PS	Plasma	Low	[236]
		Fibroblast	High	[214]
		Frontal cortex	High	[110]
		Visual cortex	High	[178]
	PI	Fibroblast	High	[214]
		Frontal cortex	High	[140]
		SN	Low	[179]
	PG	Frontal cortex	High	[110]
	APG (acyl PG)	Plasma	Low	[175]
Sphingolipid	Sphingosine	Neurons	Induce	[173]
	Ceramide	Plasma	High	[173]
		Plasma	Low	[170]
		Plasma	Low	[175]
		Frontal cortex	Low	[110]
		ACC	Low	[39]
		PVC	High	[279]
	SM	Plasma	High	[175]
		ACC	Low	[279]
		ACC	High	[279]
		PVC	High	[178]
		SN	High	[179]
	Cerebroside	SN	High	[179]
		Plasma	High	[175]
		Frontal cortex	Low	[140]
	GlcCer	Plasma	High	[217]
	GM3	Plasma	High	[217]
	GM2	Fibroblast	High	[214]
		Brain	High	[218]
	GM1	SN	Low	[213]
	Sulfatide	Frontal cortex	High	[110]
Sterol	Cholesterol	Plasma	Low	[267]
		Plasma	High	[266]
		PVC	High	[178]
Phospholipids	CP	Frontal cortex	High	[110]
	EP	Frontal cortex	High	[110]
		Plasma	Low	[175]

Polyunsaturated fats (PUFA), triacylglycerols (TAGs), docosahexaenoic acid (DHA), linoleic acid (LA), docosatetraenoic acid (DSA), eicosapentaenoic acid (EPA), osbond acid (OBA), prostaglandin (PG), phosphatidic acid (PA), phosphatidylcholine (PC), phosphatidylserine (PS), phosphatidylethanolamine (PE), phosphatidylinositol (PI), phosphatidylglycerol (PG), acylphosphatidylglycerol (APG), glucosylceramide (GluCer), gangliosides (GM), choline plasmalogen (CP), ethanolamine plasmalogen (EP), mesencephalic (MES), substantia nigra (SN), anterior cingulate cortex (ACC), and primary visual cortex (PVC).

## 10. Lipid-α-syn-Proinflammatory Cytokine Pathway in PD

The capacity of the immune system relies heavily on CNS macrophages (Mɸs) known as microglial cells [280,281]. Foamy Mɸs with lipid droplets, which store TAG and cholesterol esters (CEs), are found in various disease states, including atherosclerotic lesions tuberculosis, multiple sclerosis, certain cancers, white adipose tissue during obesity, and in bronchoalveolar lavage from individuals suffering from vaping-related lung disease [282,283,284,285]. The lipid droplets can act as energy stores, since TAG lipolysis releases fatty acids (FAs) for mitochondrial oxidation (FAO), a process that relies on long-chain FA conversion into acylcarnitines by the enzyme Cpt1a [283]. However, in Mɸs, proinflammatory signals result in diminished FAO and increased TAG synthesis with development of the lipid droplets [284]. Enhanced TAG synthesis causes Mɸs activation with increased lipid droplet formation, and massive generation of proinflammatory cytokines (e.g., IL1β, IL6, and PGE2). However, the deficiency of TAGs prevents the lipid droplets’ development and the production of proinflammatory cytokines [286]. glucosylceramide and glucosphingosine-loaded Mɸs have been directly involved in the massive generation of several proinflammatory cytokines and chemokines that lead to tissue destruction in Gaucher disease [163,164,166,167,287,288,289,290].

Microglial cell activation and proinflammatory cytokines (e.g., TNFα, IL1β, IL6, I-8, IL12, MIP1α, and IFNγ) have been severely altered in many tissues and models of PD [291,292,293,294,295,296]. The general trend observed of glycerolipids is that they are lower in PD models [272]. Both decreased and increased TAG levels and accumulated α-syn have been found in *A53*T mutants [97,98,99,100,272]. In *A53*T mutants, serum levels of TAGs and non-esterized fatty acids (NEFA) were decreased, but elevated levels of TNFα mRNA expression were found in the brain. Furthermore, expression of *Sirt2*, a gene responsible for mitigating inflammatory damage, was significantly decreased [272]. A possible mechanism for this is stimulation of hepatic lipid secretion into the serum and an increase in lipolysis. This results in increased VLDL hepatic TAGs [297]. VLDL cholesterol is secreted from the liver to supply VLDL TAGs to the body’s tissues [298]. Interestingly, lower levels of cholesterol were also found in the serum of the *A53*T mice [272]. Therefore, higher levels of TAGs and cholesterol may mitigate the pathogenesis of PD. However, as mentioned before, the effect of cholesterol intake on PD is not well understood and very controversial [235].

Recent studies in prostaglandin E2 (PGL E2) have revealed its significance in enhancing inflammation [299]. In animal models of PD, LPS stimulation and α-syn fibrils resulted in excess secretion of PGL E2 [300,301,302]. When secreted, PGL E2 will bind to EP1-4 receptors. This results in alterations of intracellular cAMP that will activate specific kinases and result in the activation of different inflammatory pathways [303]. Accordingly, increased levels of PGL E2 have been observed in the CSF of PD patients with cognitive impairments. In the CSF, levels of cytokines were also measured and found to be significantly altered. TNFα and IFNγ levels were lower, while IL1 β and IL6 levels were elevated [304]. Therefore, it appears that increased levels of PGL E2 result in cytokine alterations and neuroinflammation. Conversely, lower levels of PGL E2 have been observed in the astrocytes of DJKO mice. Although similar to the observation in the CSF of PD patients, astrocytes had decreased expression of TNFα [305]. A possible explanation for this difference lies in the different roles of the different EP receptor subtypes (EP1-EP4) [235]. In hydroxydopamine (6-OHDA)-treated mice, the knockout of EP1 and agonist of EP2 resulted in neuroprotection against neurotoxicity [235,306,307]. In the Substantia Nigra of 1-methyl-4-phenyl-1,2,3,6-tetrahydropyridine (MPTP)-treated mice, the receptor EP4 was linked to dopaminergic neuron loss and neurodegeneration [235,308]. Therefore, the role of PGL E2 and enhancing inflammation may be dependent upon which EP receptor it binds [235].

Beta-galactosylceramide (β -GalCer) and sulfatides are two sphingolipids that contribute to neuroinflammation [309]. β -GalCer is a precursor of sulfatides, and both are present in the pancreas [310] and whole blood [311]. Cytokines, such as IL1 β, IL6, TNFα, and IFNγ are known to have major influences on the pancreas [309,312]. Accordingly, altered levels of cytokines and these sphingolipids have been found in peripheral blood. When stimulated with LPS, blood cells revealed elevated levels and lower levels of β -GalCer and sulfatides, respectively. In blood cells with elevated β-GalCer, higher levels of TNFα, IL1β, IL6, IL8, and MIP1 α were detected. In blood cells with lower levels of sulfatides, TNFα, IL1β, IL6, IL8, IL12, and MIP1α were significantly lowered. In conclusion, β -GalCer increased cytokine production, and sulfatides reduced cytokine production [309]. The release of β -GalCer recruits blood leukocytes, thus contributing to inflammation [313]. Sulfatides have been recently studied for their immunomodulating effects on dendritic cells in asthma [314]. These current findings of lower levels of sulfatides contributing to neuroinflammation solidify the immunomodulatory effects found in the previous asthma study [309]. 

Increases in damage-associated molecular patterns (DAMPs), including endogenous alarm signals as well as pathogen-associated molecular patterns (PAMPs), have been recognized in many immunological cells, including dendritic cells (DCs), Mɸs, and polymorphonuclear neutrophils (PMNs), which trigger innate and adaptive immune inflammation in several diseases [315,316,317,318,319]. Two classical pathways of antigen presentation have been described for the presentation of endogenous antigens on major histocompatibility complex I (MHC class I) molecules, and the presentation of exogenous antigens, such as intracellular pathogens, on MHC class II molecules [320]. Both MHC I and MHC II are found to express on antigen-presenting cells, such as microglial cells, and dopaminergic neurons, and involved processing the antigen presentation to the surface of CD4^+^ T cells and CD8^+^ T cells [321,322]. The expression of MHC I and II can be induced in dopaminergic neurons in the substantia nigra, and α-syn-peptide-stimulated T cells have shown the development of activated subsets of helper and cytotoxic T cells and increased production of proinflammatory cytokines [322,323]. Indeed, the MHC II^+^ and MHC I^+^ microglial cells and dopaminergic neurons, as well as increased brain infiltration of effector CD4^+^ and CD8^+^ T cell subsets, have been observed in animal models and human patients with PD [18,324]. Thus, the improper metabolism of several lipids initiates the abnormal production of α-syn (Figure 1). The recognition of such α-syn by MHC I/MHC II molecules and their presentation to the corresponding T cells causes massive brain generation of several proinflammatory cytokines, which leads to neurodegeneration in PD.

In contrast to the MHC II-mediated peptide antigen processing and presentation to CD4^+^ and CD8^+^T cells [325,326,327], NKT cells are subsets of T cells that coexpress T-cell (CD3, α/βTCRs) and NKT (NK1.1)-cell surface receptors [328]. These cells recognize hydrophobic antigens presented by major histocompatibility complex class I-like molecules, such as CD1d. CD1d-restricted NKT cells are classified into two subsets, namely type I and type II. CD1d-restricted type I NKT cells express invariant T-cell receptors (TCRs) and react with lipid antigens, including the marine-sponge-derived glycolipid α-galactosylceramide. On the contrary, CD1d-restricted type II NKT cells recognize a wide variety of antigens, including glycolipids, phospholipids, sulfatides, and hydrophobic peptides, by their diverse TCRs [329,330,331,332,333,334,335,336,337,338,339,340]. Five CD1 isoforms are present in humans and are classified into two groups based on sequence similarity: CD1a, -b, and -c constitute group I, while CD1d forms group II [341]. CD1e represents an intermediate between the two CD1 groups and acts as a chaperone to facilitate lipid transfer onto CD1b and CD1d [342].

Sphingolipid activator proteins (SAPs) are ~10 kDa glycoproteins that have differential affinity for various glycosphingolipids. They have been postulated to facilitate membrane extraction and/or membrane structure of specific glycosphingolipids to enhance their degradation [343]. In addition, of the four SAPs (A, B, C, D) that derived from a common precursor, prosaposin, SAP B has a significant ability to transfer lipids to the CD1d molecule [344]. Moreover, prosaposin-deficient mice fail to develop or stimulate invariant NKT cells [345,346], suggesting that SAPs are critical for the development of NKT cells. Most of the mycobacterial-derived lipids, e.g., mycolic acid, glucose-monomycolate, phosphatidylinositol mannoside [347], lipoarabinomannan, mannosyl-β-1-phosphoisoprenoid, and mannosyl-β-1-mycoketide [348] bind to CD1 group 1. Mice express only two homologues of CD1d, i.e., CD1d1 and CD1d2 [329]. CD1d presents endogenous and exogenous lipid antigens to NKT cells in humans and mice [333,349]. The best-known subset of CD1d-restricted NKT cells uses an invariant TCRα chain (Vα14-Jα18 in mice and Vα24-Jα18 in humans), i.e., invariant NKT (iNKT) cells [350]. iNKT cells rapidly secrete IFNγ, IL4, IL17, and other cytokines upon TCR stimulation [350,351,352]. Activated iNKT cells in turn activate Mɸs and thereby impact subsequent immune inflammation [353]. These interactions highlight the critical role played by iNKT cells in bridging innate and adaptive immune responses in diseases, including cancer [354]; bacterial, viral, parasitic, and fungal infections [355,356,357]; and autoimmune diseases [358].

Several of the lipids have been found to link with the massive generation of proinflammatory cytokines (Table 3). Furthermore, upregulation of CD1d and increased infiltration of NKT cells have also been reported in PD [359,360]. Based on such observations, CD1d upregulation was postulated to be secondary to alterations in intracellular trafficking as a consequence of excess lipid accumulation. These findings potentially implicate these lipids as damage-associated molecular patterns that likely directly, and/or via CD1d-restricted binding of their presentation to NKT cells, fuel massive generation of proinflammatory cytokines that lead to neurodegeneration in PD.

## 11. Lipid-α-syn-ROS Pathway in PD

Reactive oxygen species (ROS) are highly reactive, diffusible molecules, which generate intracellularly and extracellularly for the persistence of the host defense against infectious agents and biochemical insults [361,362,363]. ROS is involved in cancer, atherosclerosis, angiogenesis, diabetes, aging, and neurological degeneration, and known to promote oxidative stress (OS) in several diseases, including age-related macular degeneration, cataracts, and uveitis [364,365,366,367,368]. Several studies have shown that lipid metabolisms contribute to the induction of oxidative damage and tissue inflammation [369]. The roles of various enzymes and lipids such as COX-2, cPLA2, ARA, and prostaglandin (PGL) have been associated with ROS generation [370]. ARA functions comprise the healing of skeletal muscle and early neurological development [371,372]. Its first contribution to inflammation was recognized in 1984, when the administration of anti-inflammatory drugs inhibited production of inflammatory PGL [373]. Certain mediators, such as calcium and cPLA2, caused the phospholipid-membrane-mediated release of ARA [374]. MTPT-induced mouse models of PD with an E-EPA diet showed increased levels of ARA, COX-2, cPLA2, and iNOS in several brain regions, including the frontal cortex and striatum, supporting the connection between ARA metabolism and the OS [110,141]. Elevated levels of α-LA have been detected in MES neurons and certain brain regions, i.e., the frontal cortex and striatum, of α-syn mouse models and human patients with PD [81,135,141]. α-LA is a major precursor for ARA [375,376]. It is therefore possible that α-LA could convert to ARA and contribute to OS-mediated neuroinflammation in PD. However, future research is needed to confirm this possibility for developing PD pathogenesis. PGL is a lipid mediator produced from ARA via COX-2 [299]. Many studies have reported both lower and higher levels of PGL E2 in PD [304,305], but in terms of overexpression of COX-2, lower levels of PGL E2 have been detected in the striatum, hippocampus, cortex, and astrocyte of mouse models [305,377]. However, reduced levels of 15-F2t-IsoP, which are the marker of OS, were observed in the striatum and cortex of mice with lower PGL E2 [377]. This contradicts the suggestion that the creation of ROS is favored over PGL E2. Therefore, further investigation into the levels of PGL E2 and OS-mediated inflammation is warranted.

Several other acidic lipids such as CL, which is present in the inner membrane of the mitochondria, have been linked to OS and the pathogenesis of PD [275,378,379,380,381]. Lower levels of CL and increased palmitate has been associated with cytochrome c release [382], oxidative stress [149], inflammation [383], and α-synucleopathies [384,385,386]. Therefore, lower CL levels seem to contribute to neuroinflammation via higher levels of α-syn and lower levels of palmitate in PD. Furthermore, cholesterol levels in male *DJ* KO mice contributed to OS [106]. The low-density lipoprotein receptor (LDLR) produces LDLs that carry cholesterol. This receptor is also a downstream target of *DJ1*. Furthermore, when stimulated with H_2_O_2_, cholesterol levels decreased in embryonic fibroblasts and astrocytes. This therefore indicates a relationship between OS and cholesterol in PD [106].

As mentioned above, DHA contributes to α-syn-mediated neuroinflammation, but a recent study introduces its role in OS-mediated neuroinflammation. In the anterior cingulate cortex of postmortem PD patients, DHA and DPA levels were elevated. To indicate OS and oxidation of tissues, F2-isoprotane levels were tested and found to be elevated [279]. This indicates that higher DHA levels induce oxidative stress. However, in PC12 cells, DHA shows a neuroprotective role against OS [387]. Therefore, the exact role of DHA in promoting oxidative stress is not well-understood.

The link between lipid metabolism and oxidative damage-induced neuroinflammation has also been observed [370]. Several of the proinflammatory cytokines, i.e., TNFα, IL1β, and IFNγ, caused cellular and mitochondrial ROS production [388,389,390,391,392,393,394,395]. Increased levels of NOS, a product of NF-κB gene expression, were observed in astrocytes [305]. Furthermore, the lipid-induced activation NF-κB pathway has been linked to inflammation by regulation of inflammasomes and stimulation of cytokines and chemokines [396]. ROS-induced oxidative damage has been observed in *DJ1*, *PINK1*, *PARKN*, *SNCA*, and *LRRK2* familial PD [397]. Several lipids are involved in the generation of such proinflammatory cytokines and ROS in mouse models and human patients with PD (Table 3 and Table 4) suggesting the involvement of the lipid- and/or α-syn-induced activation NF-κB pathway’s contribution to the generation of proinflammatory cytokines and ROS that lead to neurodegeneration in PD.

**Figure 1 ijms-23-01089-f001:**
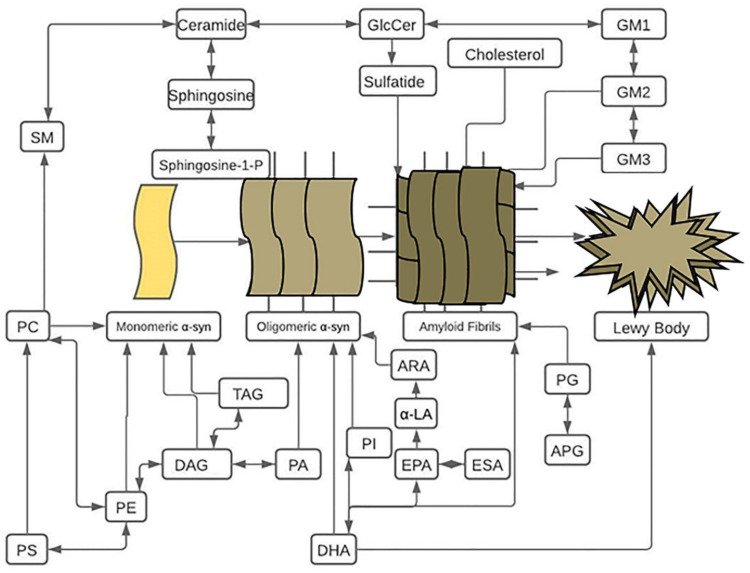
The different stages and lipids that affect alpha-synuclein (α-syn) aggregation into Lewy bodies (LBs). This chart depicts the overall relationships between lipids in this paper and their effects at different stages of LBs formation, not the exact conversions of one lipid to another. Black arrows indicate lipids that aid in the pathogenesis of PD. Phosphatidic acid (PA), phosphatidylcholine (PC), phosphatidylserine (PS), phosphatidylethanolamine (PE), phosphatidylinositol (PI), phosphatidylglycerol (PG), acylphosphatidylglycerol (APG), triacylglycerols (TAGs), diacylglycerols (DAGs), docosahexaenoic acid (DHA), α-linoleic acid (α-LA), eicosapentaenoic acid (EPA), eicosatrienoic acid (ESA), arachidonic acid (ARA), sphingomyelin (SM), glucosylceramide (GluCer), gangliosides (GM), P (phosphorus).

## 12. Conclusions

Overall, the biological function of the lipid-α-syn-cytokines-ROS pathway remains elusive; however, increasing evidence suggests that certain intracellular lipids and their impact on the induction of α-syn abnormalities, excess generation of proinflammatory cytokines, and ROS are responsible for the propagation of brain inflammation in PD (Figure 1 and Table 1, Table 2, Table 3 and Table 4). Several immune modulators, such as microglial cells and dopaminergic neuron expressions of CD1d, MHC I, and MHC II molecules, as well as increased brain infiltration of T cell subsets, including CD4^+^ T cells, CD8^+^ T cells, and NKT cells, and higher production of proinflammatory cytokines, have been observed in mouse models and human patients with PD [320,321,322,323,324]. Such T cells are known for activation of plasma B cells responsible for production of immunoglobulin G (IgG) and M (IgM) antibodies [399,400,401,402]. These antibodies are also responsible for the generation of proinflammatory cytokines in several diseases [163,287,403,404].

Interestingly, α-syn-specific antibodies and excess generation of proinflammatory cytokines have been linked to the pathogenesis of PD [405,406,407,408]. On the basis of these findings, it is suggested that microglial cells and dopaminergic-neuron-mediated cellular take-up of lipids, (e.g., TAG, GPE, PA, PE, PC, GM1, GM2, and GM3) and/or such lipid-induced alteration in pathogenic forms of α-syn (Figure 1), as well as their cellular processing and presentation by CD1d to NKT cells (for lipids) or presentation and delivery through MHC I/MHC II (for lipid-induced activated forms of α-syn) to corresponding T cell subsets (e.g., CD4^+^ and CD8^+^ T cells) cause massive generation of proinflammatory cytokines, which is lethal in PD.

Furthermore, such lipid and/or lipid-induced activated α-syn peptide processing and presentations trigger T-cell activation. These T cell subsets’ interactions with resting B cells promote B-cell differentiation into plasma cells, which could lead to the generation of α-syn-specific autoantibodies, which are all contribute to the development of chronic cellular and humoral immune inflammation, leading to neurodegeneration in PD.

Moreover, the prions, which are composed of self-propagating assemblies of a misfolded and/or aggregated α-syn, initiate the fast seeding and spreading of these defected proteins throughout the peripheral and central nervous system in several animal and human models of synucleinopathies [57,58,59,60,61,62,63,64,65,66,67]. Some of the prion diseases have shown high accumulation of such defected proteins within the secondary lymphoid tissues of the host immune system as they make their journey from the site of infection to the brain [409]. Additionally, the gastrointestinal tissues, peripheral blood mononuclear cells and brain cells of PD patients have shown higher levels of α-syn and its link to immune abnormalities [18,410,411,412,413,414,415,416]. It is therefore speculated that lipid- and cholesterol-mediated α-syn aggregation and transmission mechanistically permit several of the brain residential cells, i.e., microglial cells, neurons, and infiltrated immune cells, (e.g., T or B cells) to cause excessive processing and presentation of defected α-syn in PD. Such α-syn-mediated joint action of neurological and immunological cells fuels massive generation of proinflammatory cytokines and autoantibodies specific to α-syn, which eventually spark severe and chronic neuroinflammation and neurodegeneration in PD. However, the precise molecular mechanism(s) by which different lipids and/or such lipid-induced α-syn abnormalities might potentially lead to multiple strains of abnormal and/or prion-like α-syn, which might account for clinical heterogeneity and cause activation of cellular and humoral immune reactions that lead to the development of brain inflammation and neurodegeneration in PD, remains undetermined and needs further study. As the gold standard of treatment for PD, levodopa and its derivatives are only the standard care for PD patients [417]. However, such therapies do not alter the disease’s course, and nor do they stop or reverse the brain inflammation, which often has the greatest impact on quality of life [418,419]. Additionally, the long-term use of these drugs has been linked to several adverse effects, such as fluctuations, dyskinesia, toxicity, or loss of efficacy [420]. The relationships between these lipid and α-syn interactions open new areas of research that may identify several of the lipids, α-syn-specific antibodies and their receptors, proinflammatory cytokines and their receptors, and interesting anti-inflammatory therapies for PD and other synucleinopathies.

## Figures and Tables

**Table 3 ijms-23-01089-t003:** Lipids involved in direct activation of neuroinflammation via cytokine release.

Class of Lipid	Lipids	Mouse Models and Human PD Patients	Source and Level of Lipid	Cytokine	Source and Level of Cytokine or Inflammatory Marker (m for mRNA and *p* for Protein)	References
Glycerolipids	NEFA	*A53*T mutant mice	Serum^-^	TNF α	Brain^m+^	[272]
	TAG	*A53*T mutant mice	Serum^-^	TNFα	Brain^m+^	[272]
Eicosanoids	PGL E2	*DJ* KO mice	Astrocyte^-^	TNFα	Astrocytes^m−^	[305]
		PD patients with cognitive impairment and dementia	CSF^+^	TNFα IL1β IL6 INFγ	CSF^p−^ CSF^p+^ CSF^p+^ CSF^p−^	[304]
Sphingolipid	β -GalCer	PHA- and LPS-stimulated blood cells	PB^+^	TNFα IL1β IL6 IL8 MIP1α	PB^p+^ PB^p+^ PB^p+^ PB^p+^ PB^p+^	[309]
	Sulfatides	PHA- and LPS-stimulated blood cells	PB^+^	TNFα IL1β IL6 IL8 IL12 MIP1α	PB^p−^ PB^p-^ PB^p−^ PB^p−^ PB^p-^ PB^p-^	[309]
Sterol	Cholesterol	*A53*T mutant mice	Serum^-^	TNFα	Brain^m+^	[272]

Non-esterified fatty acids (NEFA), triacylglycerols (TAGs), prostaglandin (PGL), galactosylceramide (GalCer), Parkinson’s disease (PD), polyhydroxyalkanoates (PHA), lipopolysaccharide (LPS), cerebral spinal fluid (CSF), peripheral blood (PB), tumor necrosis factor (TNF), interleukin (IL), interferon (INF), macrophage inflammatory protein (MIP), protein expression (p), mRNA expression (m), high levels of expression (+), and low levels of expression (−).

**Table 4 ijms-23-01089-t004:** Lipids and their levels involved in oxidative stress-mediated neuroinflammation.

Class of Lipid	Lipids	Mouse Models and Human Patients with PD	Source and Level of Lipid	Reactive Oxygen Species	Source and Level of Reactive Oxygen Species	References
Eicosanoids	PGL E2	6-OHDA treated mice	Striatum^−^ Hipp^−^ Cortex^−^	15-F2t-IsoP	Striatum^p−^ Cortex^p−^	[377]
		*DJ*KO mice	Astrocyte^−^	NOS2 COX2	Astrocytes^m+^ Astrocytes^m+^	[305]
Glyercophospholipids	CL	*PINK1* KO mice	Embryonic fibroblast^−^	Palmitate	Embryonic fibroblast^−^	[275]
		*Thy1*α-syn mice	Brain^−^	n/a	n/a	[386]
Sterol	Cholesterol	*DJ1* KO	Embryonic fibroblasts^−^	H_2_O_2_	D2 Cells^+^	[273]
		*DJ1* KO	Astrocyte^−^	H_2_O_2_	D2 Cells^+^	[273]
PUFA	α-LA	MPP+ treated mice on E-EPA diet	Frontal cortex^+^	cPLA2 COX2	Striatum^m+^ Striatum^m+^	[141]
		MPP+ treated mice on E-EPA diet	Striatum^+^	cPLA2 COX2	Striatum^m+^ Striatum^m+^	[141]
	ARA	MPP+ treated mice on E-EPA diet	Frontal cortex^+^	cPLA2 COX2	Striatum^m+^ Striatum^m+^	[141]
		MPP+ treated mice on E-EPA diet	Striatum^+^	cPLA2 COX2	Striatum^m+^ Striatum^m+^	[141]
		MPTP-treated mice	Striatal synaptoneurosomes^+^	cPLA2 nNOS	Striatum^p+^ Midbrain^p+^ Striatum^p+^ Midbrain^p+^	[142]
		MPTP-treated mice	Midbrain synaptoneurosomes^+^	cPLA2 nNOS	Striatum^p+^ Midbrain^p+^ Striatum^p+^ Midbrain^p+^	[142]
	DHA	PD postmortem tissue	ACC^+^	F2-IsoP	ACC^p+^	[279]
	DPA	PD postmortem tissue	ACC^+^	F2-IsoP	ACC^p+^	[279]
Sphingolipid	GT1b	Sprague-Dawley rats	SN^−^	iNOS	SN^+^	[398]

Polyunsaturated fats (PUFA), prostaglandin (PGL), cardiolipin (CL), α-linoleic acid (α-LA), arachidonic acid (ARA), docosahexaenoic acid (DHA), docosapentaenoic acid (DHA), ganglioside GT1b (GT1b), 6-hydroxydopamine (6-OHDA), knockout (KO), alpha-synuclein (α-syn), 1-methyl-4-phenylpyridinium (MPP), 1-methyl-4-phenyl-1,2,3,6-tetrahydropyridine (MPTP), eicosapentaenoic acid (EPA), Parkinson’s disease (PD), hippocampus (Hipp), anterior cingulate cortex (ACC), substantia nigra (SN), isoprostane (IsoP), nitrous oxide (NOS), neuronal nitrous oxide (nNOS), cyclooxygenase (COX) and cytosolic phospholipase A (cPLA). Protein expression (p), mRNA expression (m), high levels of expression (+), and low levels of expression (−).

## Data Availability

Not applicable.

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
