# Peer review of "Fat and Protein Combat Triggers Immunological Weapons of Innate and Adaptive Immune Systems to Launch Neuroinflammation in Parkinson’s Disease"

_ijms, 2022, doi:10.3390/ijms23031089_

Round 1

Reviewer 1 Report

I read with interest the review by Hatton and Pandey. In this manuscript the authors review the role role of numbers of lipids and cholesterols and their connection with aSyn aggregation and induction of innate and adaptive immune reaction that trigger neuroinflammation in PD. Overall, the manuscript is well-writen and might be of potential interest for the PD & related Synucleinopathies research community.

Nevertheless, I have some concerns that the authors should address in the revised version of the manuscript.

  1. Although I understand the authors intention to make a captivating title, I would advise them to refrain from using colloquialisms such as “immunological party”. Besides being too informal and distractive, it might not be well received by potential readers, which I believe it is not the authors’ end goal.
  2. The authors practically ignored to acknowledge - there’s only one very brief sentence about it -  about one of the hottest, if not the hottest,  topic in PD field in recent years, i.e. the the prion-like cell-to-cell propagation of aSyn. Compiling evidence has shown that aSyn aggregate “seeds” can trans-synaptically spread along neuronal pathways (PMID: 34361100). Notably, aSyn pathological template seeding can start in the PNS and retrogradely propagate to the brain (PMID: 33880502; PMID: 33632316; PMID: 31254094). Further, the existence of diverse so-called aSyn “strains” is most likely responsible for the clinical heterogeneity among PD & related Synucleinopathies (PMID: 33978813; PMID: 26324905). Therefore, it would be most interesting if the authors could further comment on these breakthrough observations and speculate on: i) how the interaction of aggregate aSyn and lipids and cholesterols might impact on the prion-like spreading of aSyn pathology in PD, and ii) whether the interaction of aSyn with different lipids and cholesterols might potentially lead to multiple “strains” that might account for clinical heterogeneity.

Reviewer 2 Report

The authors provide comprehensive overview of the relation between protein alpha synuclein and lipids in respect to its aberrant behavior leading to the aggregation. The aggregation of this protein is linked to neurodegenerative diseases and thus represents the candidate for targeted medication. The submitted manuscript is very well written with logical structure. It clearly summarizes the current knowledge of the field. The synuclein protein binds to negatively charged lipids with the effect on its aggregation capacities. This reviewer suggest to briefly discuss the relationship between alpha and gamma synuclein in respect to phosphoinositides (mainly PI4,5P2). Both proteins show tendencies for the aggregation and were connected to numerous neurodegenerative diseases. In some cases these protein even colocalize (alzeheimer). Interestingly the gamma synuclein protein was recently identified as direct nuclear PI4,5P2 interactor (doi: 10.3390/cells10010068. ) see supplementary results. The authors shall briefly comment these findings in the section of the manuscript.

Reviewer 3 Report

Manuscript ID: ijms-1560988
Type of manuscript: Review
Title: Fat and Protein: A complex love story throws an immunological party in Parkinson’s disease.

This is a well-written, informative review that discusses the possible role of multiple body lipids that may induce alpha synuclein aggregation and the immune response that initiates neuroinflammation in PD. I recommend the article for publication; I have only minor comments for the authors.

Minor:

  • Introduction; please correct “niagara” to “nigra”.
  • Please standardize the spelling of some words. Authors sometimes write “triglyceride”, sometimes ”tri glyceride”, “NKT cells” or NK-T cells.
  • Page 2; Last sentences of the paragraph. When authors write about an abnormal TAG level, they should also clearly state in the text whether it is a decrease or an increase, not just refer to tables.
  • Page 4, line 6, between patients and In mice should be dot, not coma.
  • Page 4, Para 6, there is missing dot before Interestingly.
  • Page 7, “cholesterol interacts”, not “cholesterol interact”, unless it should be interacted if further you write caused.
  • Page 11, Lack of bracket after IFN γ
  • Page 11, You wrote: “A possible explanation for this difference lies in the rolls of the different EP receptors.”. Did you mean role of the different EP receptors? Please correct.
  • Page 12. Why Dopaminergic neurons is written in capital in the middle of the sentence?
  • Page 14. Last two 2-3 sentences: “α-LA could converted”. Shouldn’t be “could convert? Next sentence: “However future research needed to confirm”, shouldn’t be: “is needed”?
  • Page 15. Sentence: “This indicates DHA levels induce oxidative stress.” What levels low or high?
  • Conclusion: Sentence; “Such T cells are known for activating antibodies producing plasma B cells responsible for immunoglobulins, (e.g., IgG and IgM) production”. Should be rewritten. It is repeated twice that plasma cells produce antibodies/immunoglobulins in short sentence. The last sentence of the discussion: what findings and what studies? Please precise, otherwise the reader doesn't know what it's about.

Round 2

Reviewer 1 Report

The authors have successfully addressed my concerns, therefore I fully recommend the revised version of the manuscript for publication.